# Assessing the Impact of Urban Lifestyle and Consumption Values on Conversion Intention: A Study towards Energy Sustainability

**Hilda Hilmiyati-Mas'adah** [1,2]🆔, **Achmad Sudiro** [2], **Fatchur Rohman** [2], **Agung Yuniarinto** [2],
**Dzikri Firmansyah Hakam** [3,*]🆔 **and Herry Nugraha** [1]

1   PT Perusahaan Listrik Negara (Persero), Jakarta 12160, Indonesia; hilda09@student.ub.ac.id (H.H.-M.);
    herry.nugraha@pln.co.id (H.N.)
2   Faculty of Economics and Business, Brawijaya University, Jakarta 12870, Indonesia;
    achmad_sudiro@ub.ac.id (A.S.); fatchur@ub.ac.id (F.R.); agung@ub.ac.id (A.Y.)
3   School of Business and Management, Bandung 40132, Indonesia
*   Correspondence: dzikri.hakam@sbm-itb.ac.id

**Abstract:** Energy converter innovation has shifted the world's cooking energy from wood, coal, kerosene, and liquid petroleum gas (LPG) to electricity. This paper identifies the factors influencing customers' intention to switch their cooking energy from LPG to electricity. The study proposes a conversion intention (COIN) framework with urban convenience and perceived alternative value (PAV) as predictor variables and examines the mooring effect of conversion cost to COIN. The data were gathered by a cross-sectional survey, and the PLS-SEM approach was applied to 194 LPG users in Jakarta, Indonesia. The results reveal that PAV mainly determines cooking energy conversion and partially mediates the relationship between urban convenience and COIN. Conversion cost indicates no significant moderating effect of PAV on COIN. By conducting this study, we contribute to the literature by integrating the theory of consumption value (TCV) with the value-based adoption model (VAM), generating the indicators of urban convenience based on time-oriented advertisement categories and applying the consumption values of the TCV as the dimensions of PAV on the VAM framework. The findings of the paper provide a more in-depth understanding of customers' motivations when switching from LPG to electric power, particularly for household cooking energy, as well as opportunities for the government and electronic manufacturers to promote more sustainable energy consumption patterns.

**Keywords:** intention; value-based model; consumption value; household energy; urban; PLS-SEM

## 1. Introduction

Accessibility to energy and the intensity of energy consumption are indirect indicators of a country's welfare and economic development. Energy consumption drives industry, business, transportation, building, and household activities. The household sector reaches 25% of the world's total energy consumption [1]. The governments of developing countries strive for the equitable distribution of community welfare through energy subsidies as an economic stimulus. Controlling the selling price of LPG and electricity at an affordable level is expected to encourage the increased consumption of products and energy.

Energy diversification, which is carried out in several countries such as Nepal [2] and Indonesia, is expected to reduce peoples' dependence on LPG, which greatly burdens LPG import spending [3]. Indonesia is an emerging middle-income country [4] with a household energy consumption level that is roughly 10.1% of that of China. Electricity accounts for 24.4% of total energy consumption, which is equivalent to 374.46 kWh per capita [5,6]. The subsidized LPG expenditure reached USD 9.11 billion, and electricity reached USD 4.03 billion for the 2022 outlook [7], although the accumulated energy subsidies from 2015 to 2020 were withheld by 20% of that of the previous period (2008–2014). Meanwhile,

maximizing the use of natural resources available in Indonesia would increase national energy stability [8].

Indonesia ranks tenth in the world's electricity consumption, and annually tends to increase. During the 2020 crisis, the demand of household consumers for upgrading to an installed power capacity of 1300 VA or higher showed an upward trend (Figure 1) compared to the lower capacities, resulting from growing dependency on electric equipment in daily life. Household electricity consumption continues to rise, particularly for living conveniences such as air conditioners (almost 10% used in Indonesia [9]), water pumps, lights, TVs, refrigerators, entertainment equipment, and other electronic equipment [10]. Meanwhile, the electricity reserve margin in the Java-Bali grid, including the 35 GW megaproject, reaches an average of 55.55% [11]. Although the electricity supply is sufficient to meet the increasing demand for electricity and there are more varied choices of electric kitchen appliances on the market, the trend of LPG consumption is continuing to increase (Appendix A, Figure A1). The problem is that people are less enthusiastic about the intention to switch from LPG to electric power, particularly for household cooking energy.

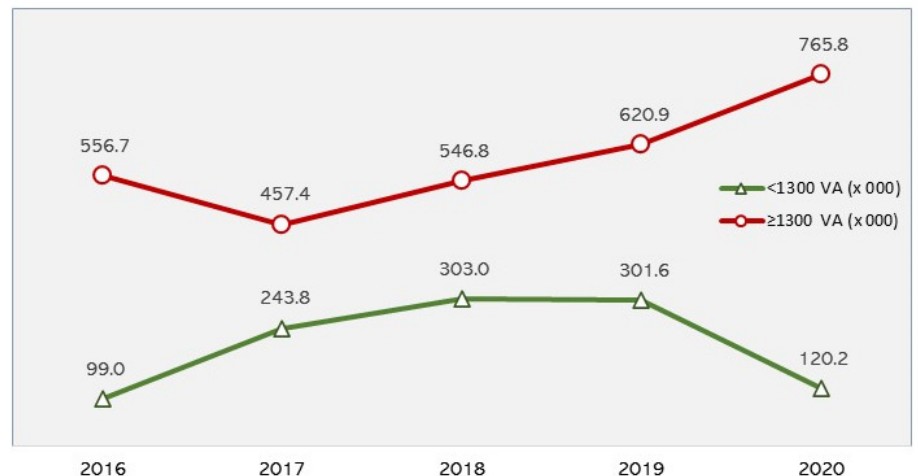

**Figure 1.** Demand for upgrading installed capacity for household electricity from customers in Indonesia from 2016–2020. Source: customer transaction database AP2TL of PT PLN (Persero) accessed on 6 May 2021 (produced by Author).

A conversion program is expected to strengthen national energy sovereignty. However, shifting from LPG-fueled to electric power for household cooking energy has not been as successful as expected [12–14]. Recently, Indonesia ran a pilot project to find people's responses to using induction stoves to convert LPG to electric power as their daily cooking energy. The program targeted a lower installed capacity of 450–900 VA. In order to reduce the switching barrier and attract customers for switching, every participant was offered a free induction stove and up-grading electricity capacity. The program was postponed for further evaluation of effectiveness issues. In the present study, we examined households with higher installed capacities.

Consumer intention is frequently used to estimate actual consumer action [15,16]. Previous research on energy consumption discussed green energy [17], energy efficiency [18–20], smart homes [21,22], and energy transformation, as summarized by [2]. This paper would be one of few quantitative pieces of research that explore conversion intention regarding cooking energy.

According to Le Meunier-FitzHugh, service-dominant logic provides an integrated solution for customers focused on service and customer orientation [23]. In the context of electricity consumption, the value propositions of power-generating companies and electric kitchen appliance (EKA) manufacturers, which offer better benefits for consumers than LPG-fueled cooking appliances, are determining factors that contribute to motivating the integration of resources by consumers and other actors who collaborate in value co-creation while consumers are using the product [24]. The decision to choose an alternative

product is influenced by perceptions that are built from one or more consumption value combination [25] because the value of a product is unique depending on who would receive the benefits [26]. The value-based adoption model is one of the research frameworks frequently used to investigate the impact of benefits on conversion intention to a new private-owned product [27], such as electric kitchen appliances.

A few concepts of convenience have been introduced by Brown [28], Berry [29], and Lotz [30]. Previous studies used the term "convenience" as an independent variable based on the concepts of Brown [31,32], Berry [33,34], or Lotz [35], with the operational definitions that were related to the topics studied. To the best of the researcher's knowledge, convenience based on the time-oriented advertisement content categories of Gross and Sheth's work [36] is not widely discussed.

This study aims to explore the factors influencing the intention of LPG consumers to switch to electricity as their primary cooking energy source. The research is motivated by the need to understand the opportunities for energy policy and industry stakeholders to promote more sustainable energy consumption patterns in Indonesia. Overall, this research contributes to the understanding of the factors influencing conversion intention in the context of LPG consumption in Jakarta, with implications for the government (in delivering energy diversification programs) and industry stakeholders.

The study is novel in that it proposes a conversion intention framework via the integration of the theory of consumption value and the value-based adoption model, which involve two key factors that influence conversion intention: urban convenience developed from content analysis categories in time-oriented advertisements [36] and perceived alternative value (PAV) based on the theory of consumption values. Additionally, the research investigates the moderating effect of conversion costs on conversion intention (COIN) [32].

The paper is structured into six sections. Section 1 provides an introduction to the research problem and the objectives of the study. It also highlights the significance of the study and the research questions that will be addressed. Section 2 presents the literature review. Section 3 describes the methods used in the study. This includes the research design, sampling, and data collection and data analysis techniques. Section 4 presents the results of the study. It includes findings related to the research objectives and provides a detailed analysis of the data collected. Section 5 provides a discussion of the results and their implications. Section 6 offers a conclusion that summarizes the main findings of the study and highlights their implications for policy and practice. It also includes recommendations for future research and conclusions on the research questions addressed in the study.

## 2. Literature Review

The advanced technology innovation of energy converters has also penetrated electric kitchen appliances, such as electric ovens, microwave ovens, toasters, induction cookers, air fryers, etc. As a result, electronics manufacturers are continuously and fiercely attempting to retain and attract consumers by offering the latest superior features. In the adoption process, people are forced (or voluntarily) to change (permanently or temporarily) the way they act from their previous comfort behavior [37]. Intentions are often used as a measure to estimate the actual action of consumers [16]. Some established theories explored switching/adoption intention or behavior, such as push-pull mooring, which analyzes factors from both previous and new systems [38], technology acceptance models [39], UTAUT [40], value-based adoption models [41], and protection motivation theory [42]. Adoption Intention is not only influenced by external factors, such as economic capacity, gender, consumption patterns, infrastructure, and resource availability [43,44]. Intention is influenced by perceived value as an intervening variable of the benefits (usefulness and enjoyment) and sacrifices (technicality and fees), as seen in a previous study using the value-based adoption model (VAM) framework [27,45].

In this study, conversion intention describes a person's action tendency to change their behavior from initial conditions to new conditions with monetary and/or nonmonetary

consequences [37]. Intention is a predictor of switching behavior to stop using existing products and/or switch to alternative products [46].

In general, innovations are developed for a definite purpose to fulfill what is desired [47]. Product invention has changed urban lifestyles and vice versa. Urbanization has changed peoples' lifestyles to be more concerned about time. The convenience customers perceive while using a product is an attractive advantage [32,34] regarding an individual's belief as a result of a subjective assessment of the benefits. Therefore, time-oriented factors often serve as the primary appeal of a product's benefit.

Urban convenience describes the degree to which the individual perceives the time-oriented convenience related to using electric kitchen appliances [36]. It is an extrinsic motivation of a customer's decision process to obtain any benefits [48]. Previously, Brown proposed five dimensions of convenience to reduce and clarify measurement ambiguity: time, place, acquisition, use, and execution [32], as shown in Table 1. Meanwhile, Berry emphasizes the idea of service convenience, which consists of the following elements: decision, access, transaction, benefit, and postbenefit [34]. This study follows applications in magazine advertisements based on content categorization to obtain the variable indicators of convenience and more accurately measure time-oriented urban lifestyles. The items were self-developed to describe time-oriented appeals: being fast to use, saving time, viewing time as scarce or valuable, specifying the amount of time required, making double use of time, and escaping a typically time-oriented lifestyle [36]. Urban convenience represents a benefit factor suited to urban lifestyles in which time is a scarce resource. Previous studies found that benefits have a significant influence on intentions [34,49,50]. Nonetheless, some studies produce insignificant results [49,51,52]. According to the explanation above, there are some research gaps that need to be filled.

**Table 1.** References of convenience dimensions.

| Context | Convenience Dimension | Results | Source |
|---|---|---|---|
| Convenience concept framework | Time, place, acquisition, use, execution | Convenience is a continuum that influences product marketing strategy | [28] |
| Magazine advertisement appeals | Fast to use, saving time, time as scarce/valuable, amount of time required, double use of time, escaping a typically time-oriented lifestyle | Time-oriented advertising in terms of portion and main attraction increases every year | [36] |
| Concept of service convenience | Decision, access, transaction, benefit, postbenefit | Service convenience has a significant impact on service evaluation | [29] |
| Examine the effect of convenience wireless LAN on intention in the extended TAM framework | Time, place, execution [28] | Perceived convenience influences intention indirectly through perceived usefulness | [31] |
| Develop and examine the factors that influence online search and shopping intentions | Overall speed of process, ease of finding what i want, time savings, instant ability to get items, freedom from hassles | Convenience does not significantly affect intention | [30] |
| Testing the effect of perceived convenience on mobile learning acceptance | Items adopted from [31] | Perceived convenience has no effect on perceived usefulness, perceived ease of use, and attitude using mobile learning | [53] |
| Examine the factors that influence the intention to use the e-reader | Items adopted from [28] | Convenience affects intention | [32] |
| Testing the convenience effect of online booking on intention to book highly complex travel package | 3 item adopt from [30] | The attitude of ordering online package based on individual preference is unaffected by convenience | [35] |
| Examining service convenience as a factor influencing cruise ship passengers' satisfaction and intent to return. | Decision, access, transaction, benefit, postbenefit [29] | Service convenience has a significant effect on satisfaction and revisit intention | [34] |
| Examine low-budget fitness center customer loyalty. | Decision, access, transaction, benefit, postbenefit [29] | perceived quality, service convenience affects perceived value, satisfaction and loyalty | [33] |
| Experimental research on the queuing effect on the buying decision. | Length of queue (transaction) | Long queues increase the number of purchases | [51] |
| Comparing the effect of convenience on loyalty and perceived value at department stores and supermarkets | Decision, access, transaction, benefit, postbenefit [29] | Access and postbenefit in department stores, as well as access and benefits in supermarkets, have no significant effect on perceived value | [54] |
| Examining the effect of convenience on the hedonic and utilitarian value of products that affect the intention to adopt mobile banking. | Access, transaction, benefit, postbenefit [29], search, evaluation | Because banking products are relatively similar, search and evaluation have no effect on the intention to adopt mobile banking | [52] |

The value-based adoption model framework is frequently used to investigate the conversion intention of a new, privately owned product [27]. In VAM theory, Zeithaml's idea influences the concept of "perceived value," which is defined as a combination of positive values labeled as benefits and negative values classified as sacrifices [55]. Another perspective on the value that is involved in the customer decision-making process is the consumption value of TCV. In this study, "perceived alternative value" is defined as the value of electric kitchen appliances perceived by individuals as alternative products that drive consumption decisions [25]. It is formed by five dimensions of consumption values: functional, social, emotional, conditional, and epistemic value. Previous research has found that perceived alternative value has a significant impact on conversion intention [41,56] but only partially on certain segments [57]. Further, the benefit has a significant influence on the perceived alternative value [33,54].

Besides the disadvantages of a current product, which gives push effects to conversion intention, mooring factors might exist as an inhibitor. Conversion costs are defined as the monetary and nonmonetary preliminary costs [58] of converting from LPG-fueled electric kitchen appliances, including economic risk costs [59], evaluation costs [60], learning costs, and set-up costs [61]. Previous research has identified financial incentives as an amplifying factor that strengthens the effect of perceived alternative value on conversion intention [56], although conversion costs are positioned as a mooring factor for switching [32] or re-purchasing [59] intentions. However, there is no evidence that conversion costs inhibit intentions [56].

The findings and outcomes of this research will provide some theoretical and practical contributions to understanding the influential motives that are considered by customers through reducing LPG consumption, increasing electricity for cooking energy, considering electricity as the primary household energy, a willingness to switch to LPG [62], and a desire to switch to electricity as their cooking energy [63]. Based on the literature review, this study proposes the framework presented in Figure 2.

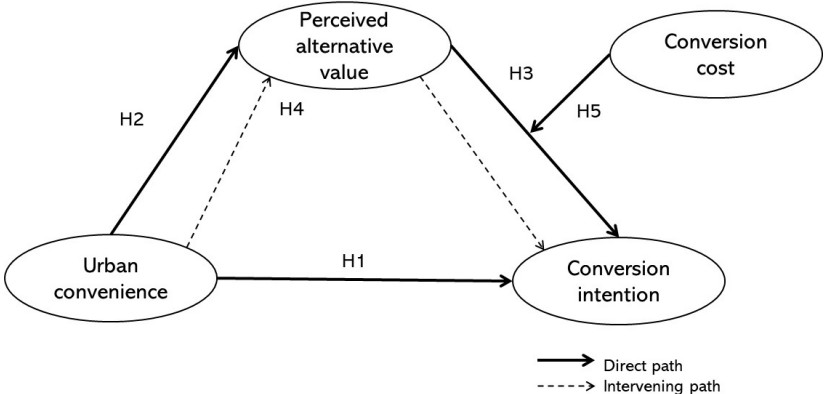

**Figure 2.** Proposed research framework.

Therefore, the following assumptions were developed:

**H1.** *Urban convenience has a significant impact on conversion intention;*

**H2.** *Urban convenience has a significant impact on perceived alternative value;*

**H3.** *Perceived alternative value has a significant impact on conversion intention;*

**H4.** *Perceived alternative value mediates the relationship between urban convenience and intention to convert LPG to electricity as cooking energy;*

**H5.** *Conversion cost has a significant impact on conversion intention.*

### 3. Data and Methods

This study applied a quantitative research methodology. Multivariate analysis was applied to test the hypotheses and models constructed using three or more variables [64] using a partial least squares-structural equation-modeling (PLS-SEM) approach to predict the value of the endogenous variables [65]. The data were collected primarily through an online survey of 194 LPG users (62.4% women and 37.6% men). According to Marcoulides et al. [66], the minimum sample size in applying PLS-SEM is 10 times the highest number of formative indicators, equal to 120. Therefore, the respondents who participated in this cross-sectional survey (held from December 2021 to March 2022) are adequate samples. With regards to age, they were two groups: 15–35 years (26.3%) and 35 years or above (73.7%). The demographic information and energy consumption patterns (Appendix A, Table A1) are included in the questionnaire so as to better understand consumer behavior (Table 2). As the study relates to daily activity for cooking at home, 51.9% of respondents had flexible time (22.7% worked from home, 7.7% entrepreneur, 7.7% retired, 7% doing domestic tasks, 2.1% others) and 47.9% had fixed time (40.7% worked in the office, and 7.2% were students). With regards to usage experience, the majority never used electric kitchen appliances (induction stove and/or air fryer). In order to enrich the interpretation of the quantitative data, some respondents were selected to be interviewed regarding the research variables to deepen the intent behind the responses.

**Table 2.** Energy consumption characteristics of respondents.

| Characteristics | Frequency | Percentage (%) |
|---|---|---|
| Installed power | | |
| 1300 Volt Ampere (VA) | 55 | 28.35 |
| 2200 Volt Ampere (VA) | 82 | 42.27 |
| 3500 Volt Ampere (VA) | 30 | 15.46 |
| 4400 Volt Ampere (VA) | 13 | 6.70 |
| 5500 Volt Ampere (VA) and above | 14 | 7.22 |
| LPG consumption | | |
| Subsidized LPG (3 kg gas cyl.) | 52 | 26.80 |
| Nonsubsidized LPG | 108 | 55.67 |
| Both | 34 | 17.53 |
| Electric kitchen appliance utilization | | |
| Induction stove | 18 | 9.28 |
| Air fryer | 21 | 10.82 |
| Both | 14 | 7.22 |
| None of them | 141 | 72.68 |
| Electric kitchen appliance evaluation | | |
| Induction stove | 134 | 69.07 |
| Air fryer | 60 | 30.93 |
| Technology Familiarity | | |
| User | 42 | 21.65 |
| Nonuser | 152 | 78.35 |

This survey applied purposive convenience sampling in terms of household electricity customers in Jakarta and the extended urban area, including Bogor, Depok, Tangerang, and Bekasi (Jabodetabek), whose installed capacity was 1300 Volt Ampere (VA) or above and who partly or fully used LPG-fueled stoves for cooking. The five-point Likert-type scale questionnaires were adapted from the instruments of previous researchers (Table 3) regarding conversion intention under six items [62,63]. In order to measure the urban convenience of electric kitchen appliances, six time-oriented appeal categories of advertisement content were used [36].

**Table 3.** Descriptions, means, and standard distributions of reflective measures of constructs.

| Variable | Item | Description | Ref | Mean | SD |
|---|---|---|---|---|---|
| Urban Convenience (UC) | UC.1 | EKA is fast to use | | 3.89 | 0.88 |
| | UC.2 | EKA is a means of saving time | [36] | 3.61 | 0.95 |
| | UC.3 | I'm busy and regard time as scarce/valuable | | 3.59 | 0.99 |
| | UC.4 | The amount of time required to use EKA is specified | | 3.70 | 0.90 |
| | UC.5 | I can use EKA while doing other activities | | 3.82 | 0.90 |
| | UC.6 | EKA gives me more time to relax | | 3.60 | 0.93 |
| Conversion Intention (COIN) | CI.1 | I prefer to think of electricity as my primary source of cooking energy. | [62] | 3.35 | 1.02 |
| | CI.2 | I'm thinking about boosting my electricity consumption as my cooking energy. | | 3.27 | 1.04 |
| | CI.3 | I'm thinking about reducing my LPG consumption as my cooking energy source. | | 3.24 | 1.00 |
| | CI.4 | I am determined to switch to EKA usage | | 3.16 | 1.00 |
| | CI.5 | The chance of my switching to EKA is high | [63] | 3.24 | 1.02 |

The service-dominant logic (SDL) concept underlying the selection of the PAV indicators states that a product value proposition that brings benefits to consumers would be considered in deciding on alternative products [24]. Therefore, this study applied a different approach to the VAM benefits and sacrifices trade-off; PAV constitutes five dimensions of consumption values [25], four items for the functional values, and two items for each social value, conditional value [67], emotional value, and epistemic value [68], as shown in Table 4.

**Table 4.** Descriptions, means, and standard distributions of formative measures of constructs.

| Variable | Dimension | Item | Description | Ref | Mean | SD |
|---|---|---|---|---|---|---|
| Perceived Alternative Value (PAV) | Functional value | FV.1 | The foods prepared by an EKA have consistent quality | [67] | 3.59 | 0.94 |
| | | FV.2 | The EKA is durable | | 3.51 | 0.84 |
| | | FV.3 | The use of EKA would save my cooking cost | | 3.17 | 1.01 |
| | | FV.4 | EKA is reasonably priced | | 3.31 | 1.01 |
| | Social value | SV.1 | Using EKA would improve the way that I am perceived | [67] | 3.25 | 1.10 |
| | | SV.2 | Using EKA would make a good impression on other people | | 3.28 | 1.09 |
| | Emotional value | EMV.1 | Using EKA instead of LPG stove would feel like the morally right thing to do | [68] | 3.07 | 1.09 |
| | | EMV.2 | Using EKA instead of LPG stove would make me feel like a better person | | 3.09 | 1.03 |
| | Conditional value | CV.1 | I would use EKA instead of LPG stove if there were a subsidy for electricity bills | [67] | 3.72 | 1.13 |
| | | CV.2 | I would use EKA instead of an LPG stove if electric power is available all the time | | 3.88 | 0.99 |
| | Epistemic value | EPV.1 | I like to search for the new different product information | [68] | 3.90 | 0.86 |
| | | EPV.2 | I am willing to seek out novel information about EKA | | 3.83 | 0.94 |
| Overall perceived alternative value | | | Overall, EKAs deliver me good value | [56] | 3.58 | 0.87 |
| Conversion cost (CC) | Economic risk cost | RC.1 | I worry that EKA won't work as well as expected | [61] | 3.26 | 1.09 |
| | | RC.2 | Switching to electric power as cooking energy causes monetary cost causes monetary cost | [59] | 3.86 | 0.95 |
| | | RC.3 | There is a certain benefit (subsidized LPG) I would not retain if I were to switch cooking energy | [69] | 3.14 | 1.08 |
| | Evaluation cost | EC.1 | Comparing the benefits of LPG-fueled kitchen appliances to those of EKAs takes too much time and effort, even when I have the information | [61] | 2.94 | 1.03 |
| | | EC.2 | I don't have the time or energy to thoroughly evaluate a variety of EKAs | [60] | 2.79 | 1.05 |

**Table 4.** *Cont.*

| Variable | Dimension | Item | Description | Ref | Mean | SD |
|---|---|---|---|---|---|---|
| | Learning cost | LC.1 | Learning to use the features offered by EKAs as well as using my LPG-fueled kitchen appliance would take time | [61] | 3.02 | 1.03 |
| | | LC.2 | Even after switching, it would take effort to get up to speed with an EKA | | 3.35 | 0.97 |
| | Set-up cost | SC.1 | When switching to electric power for cooking energy, there are numerous formal procedures to upgrade installed power capacity | [59, 61] | 2.76 | 1.12 |
| | | SC.2 | The process of switching to EKA takes time | [61] | 3.17 | 1.03 |
| Overall conversion cost | | | Switching from LPG-fueled to electric kitchen appliances will incur much loss for me | [70] | 2.94 | 1.00 |

EKA = electric kitchen appliance (induction stove or air fryer as options for evaluation). t-statistic > t critical 1.960; *p*-value < 0.05.

The data processing and hypotheses were analyzed with PLS-SEM, and we statistically predicted the output through Smart PLS procedures [71]. A total of 11 items for the reflective scales and 21 items for the formative scales met the instrument's validity and reliability test. The measurement model of the questionnaire was separately assessed because the reflective and formative scales were based on different concepts [72].

## 4. Results

### 4.1. Measurement Model Assessment

4.1.1. Reflective Measurement Model Assessment

Four analytical procedures were applied to the reflectively measured constructs: (1) assessment of item loadings; (2) internal consistency reliability (Cronbach's alpha, $\rho_A$, and composite reliabilities); (3) convergent validity (average variance extracted/AVE), and (4) discriminant validity (Hetero Trait Mono Trait/HTMT) [72]. The results in Table 5 confirm that all outer loadings of the items are above the threshold of 0.708 (Hair et al., 2021), so all items take part in the structural model assessment. The higher internal consistency reliability value indicates a better level of reliability but does not exceed 0.95 for the redundancy indications [73]. Urban convenience and conversion intention both show a Cronbach alpha (0.924 and 0.932) and composite reliability (0.940 and 0.949) of less than the threshold. The real reliability ($\rho_A$), as proposed by Dijkstra-Henseler [73], lies between the extreme value of Cronbach's alpha, without the indicator's weight and composite reliability ($\rho c$), which takes into account the weight of each indicator. Additionally, the AVE values (0.725 and 0.788) are above the threshold of 0.5, which evidences the attainment of convergent validity [74]. Discriminant validity, which statistically compares two different latent variables, indicated a Heterotrait-Monotrait (HTMT) value of 0.678, which is less than the threshold of 0.9 [73]. This explains the appropriateness of the reflective measurement instrument.

**Table 5.** Validity and reliability test of reflective measures constructs.

| Variable | Item | Outer Loading | Cronbach Alpha | Dijkstra-Henseler Reliability ($\rho$A) | Composite Reliability ($\rho$c) | Average Variance Extracted (AVE) |
|---|---|---|---|---|---|---|
| Urban Convenience | UC.1 | 0.774 | 0.924 | 0.925 | 0.940 | 0.725 |
| | UC.2 | 0.866 | | | | |
| | UC.3 | 0.880 | | | | |
| | UC.4 | 0.892 | | | | |
| | UC.5 | 0.820 | | | | |
| | UC.6 | 0.871 | | | | |
| Conversion intention | CI.1 | 0.822 | 0.932 | 0.939 | 0.949 | 0.788 |
| | CI.2 | 0.915 | | | | |
| | CI.3 | 0.820 | | | | |
| | CI.4 | 0.946 | | | | |
| | CI.5 | 0.929 | | | | |

4.1.2. Formative Measurement Model Assessment

In contrast to reflective indicators, which reflect latent variables, formatively measured indicators generate latent variables. There is no interchangeable issue between the indicators, so it is not necessary to test the reliability of the indicators, the internal consistency reliability, or the discriminant validity [75]. Three analytical procedures for assessing the PAV and CC indicators were applied: (1) convergent validity; (2) collinearity; and (3) the significance and relevance of the indicators [72]. Convergent validity was assessed by correlating the formative indicators with a single reflective global indicator. As depicted in Table 6, the path coefficients of PAV (PAV → overall perceived alternative value, β = 0.838) and CC (CC → overall conversion cost, β = 0.768) are higher than 0.708 [72], thus implying that convergent validity was attained by both of them. The collinearity assessment of the formative indicators ensured that there were no strong correlations between the construct indicators, indicated by an outer VIF of less than 10 [76]. Simply removing the indicators that are strongly correlated will have an impact on those models with a weak theoretical basis [77]. Table 6 shows that the outer VIF of PAV (1.93–5.15) and CC (1.39–2.61) are all below the threshold.

**Table 6.** Redundancy analysis structural model estimates.

| Correlation | Path Coefficient (β) | t Statistics | *p*-Value |
|---|---|---|---|
| Perceived alternative value → Overall Perceived alternative value | 0.838 | 39.570 | 0.000 |
| Conversion cost → Overall conversion cost | 0.768 | 20.909 | 0.000 |

Notes: path coef. > 0.70; t-values tested 1.96 at sig. level of 5% [72].

The weight of an indicator shows its relative role in its construct. The more indicators in a construct, the smaller the average contribution of the indicators, which makes it more difficult to obtain a statistically significant weight. Per default, we needed to use 10,000 bootstrap subsamples using BCA Bootstrap and a two-tailed test at a significance level of 5%. The results are shown in Table 7. In general, most of the outer weights of the indicators are not significant, except FV.4 of PAV (t = 3.388), RC.1 (t = 2.446), and SC.1 (t = 2.409). The indicators that are statistically insignificant in weight were further analyzed for their absolute importance via a loading value that shows a t-value above its critical point to then be retained in the model [74]. Thus, all the indicators of PAV were retained in the formatively measured constructs, even though not every indicator's weight was significant. Indicators RC.3 and LC.2 of CC show a t-value (of weight and loading) below the suggested level of 1.960, implying that these indicators did not make a relative and absolute contribution to their construct.

**Table 7.** Collinearity, significance, and relevance testing of formatively measured constructs.

| Construct | Item | Variance Inflation Factor (VIF) | t-Statistics | *p*-Value (Weight) | Loading | t-Statistics | *p*-Value (Loading) |
|---|---|---|---|---|---|---|---|
| Perceived alternative value | FV.1 | 2.18 | 0.675 | 0.496 | 0.698 | 10.688 | 0.000 |
| | FV.2 | 2.06 | 0.392 | 0.696 | 0.665 | 6.802 | 0.000 |
| | FV.3 | 2.57 | 1.095 | 0.279 | 0.756 | 11.510 | 0.000 |
| | FV.4 | 1.95 | 3.388 | 0.001 | 0.768 | 13.386 | 0.000 |
| | SV.1 | 4.64 | 0.284 | 0.776 | 0.535 | 6.178 | 0.000 |
| | SV.2 | 5.15 | 0.206 | 0.837 | 0.592 | 7.404 | 0.000 |
| | EMV.1 | 3.06 | 1.261 | 0.207 | 0.781 | 13.085 | 0.000 |
| | EMV.2 | 3.69 | 1.420 | 0.147 | 0.798 | 12.650 | 0.000 |
| | CV.1 | 1.93 | 0.873 | 0.390 | 0.318 | 3.191 | 0.001 |
| | CV.2 | 2.63 | 1.559 | 0.123 | 0.600 | 8.672 | 0.000 |
| | EPV.1 | 2.09 | 1.490 | 0.133 | 0.573 | 6.676 | 0.000 |
| | EPV.2 | 2.64 | 0.682 | 0.491 | 0.673 | 8.562 | 0.000 |
| Conversion cost | RC.1 | 2.13 | 2.446 | 0.016 | 0.771 | 7.148 | 0.000 |
| | RC.2 | 1.75 | 0.167 | 0.868 | 0.420 | 2.836 | 0.004 |
| | RC.3 | 1.39 | 0.911 | 0.374 | 0.235 | 1.499 | 0.138 * |
| | EC.1 | 1.81 | 0.999 | 0.318 | 0.709 | 6.147 | 0.000 |
| | EC.2 | 2.61 | 1.419 | 0.153 | 0.827 | 9.282 | 0.000 |
| | LC.1 | 1.86 | 0.978 | 0.330 | 0.515 | 4.117 | 0.000 |
| | LC.2 | 1.96 | 1.362 | 0.170 | 0.175 | 1.120 | 0.261 * |
| | SC.1 | 1.64 | 2.409 | 0.015 | 0.682 | 5.765 | 0.000 |
| | SC.2 | 2.14 | 0.049 | 0.961 | 0.405 | 2.830 | 0.005 |

\* Insignificant, deleted items.

### 4.2. Structural Model Assessment

A structural model assessment was carried out by only involved the indicators that passed the evaluation stage:

(1) Collinearity testing. The inner VIF was used as a measure of collinearity, with a critical value limit that did not appear to have a strong correlation between the constructs in the structural model (as high as 5) [72]. Table 8 shows that the structural model gives VIF values in the range of 1.35–3.19 at a tolerance level of 5%. Therefore, it can be concluded that there is no collinearity issue between the constructs in the structural model;

**Table 8.** Structural model collinearity test.

| Path | Inner VIF |
|---|---|
| Urban convenience → Conversion intention | 1.457 |
| Perceived alternative value → Conversion intention | 1.459 |
| Conversion cost → Conversion intention | 1.317 |

(2) Testing the significance and relevance of the relationship. The strength of a correlation is indicated by the path coefficient value, while the significance of the relationship between exogenous constructs and endogenous constructs is based on the direct effect. The result shown in Table 9 show that all the hypotheses are accepted except for hypothesis 5, for which the t statistics < 1960 and the *p*-value > 0.05. As shown in Tables 3 and 4, the simplicity of operating procedures and the automation features embedded in EKA are the strongest convenience that encourage the intention to increase a user's electricity consumption for cooking energy. It can be concluded that the effect of the perceived alternative value on conversion intention (H3: β = 0.549) is greater than urban convenience on conversion intention (H1: β = 0.352). The perceptions of consumption values that are expected to

be fulfilled by consuming products are extremely important in motivating action. In addition, the total effect of perceived alternative value ($\beta$ = 0.632) in partially mediating urban convenience to conversion intention is the sum of the direct effect ($\beta$ = 0.352) and the mediating effect ($\beta$ = 0.280). It means that perceived alternative value is positively able to increase the impact of urban convenience on conversion intention;

**Table 9.** Structural model estimates.

| Path | Path Coefficient ($\beta$) | t Statistics | Result |
|---|---|---|---|
| Urban convenience → Conversion intention | 0.352 | 5.300 | H1 accepted |
| Urban convenience → Perceived alternative value | 0.511 | 8.442 | H2 accepted |
| Perceived alternative value → Conversion intention | 0.549 | 8.040 | H3 accepted |
| Urban convenience → Perceived alternative value → Conversion intention | 0.280 | 5.098 | H4 accepted |
| Conversion cost × Perceived alternative value → Conversion intention | −0.039 | 0.922 | H5 rejected |

Notes: t-values test: 1.96 at a sig. level of 5%.

(3) Explanatory power testing to measure the degree of empirical data could be explained by model equations through the endogenous variable coefficient of the determinant ($R^2$) and effect size ($f^2$). The result in Table 10 shows that the $R^2$ of PAV is 26.1%. Meanwhile, the impact of the moderating variable of firming model estimation of conversion intention ranges from 62.3% to 64.6%. As a result, the moderation variable's size effect is quite small (6.4%);

**Table 10.** Results of $R^2$ and the model's explanatory power.

| Construct | $R^2$ Excl. Moderating Effect | $R^2$ Incl. Moderating Effect | Effect Size ($f^2$) |
|---|---|---|---|
| Perceived alternative value | 26.1% | 26.1% | |
| Conversion intention | 62.3% | 64.6% | 6.4% |

$f^2$ 0.020–0.150: low; 0.150–0.350: medium; ≥0.350: high [74]; For social science research: $R^2$ = 0.75 high; $R^2$ = 0.5 medium; $R^2$ = 0.25 low [72].

(4) In order to estimate the condition that, sometimes, consumers react unpredictably in response to marketing activities, we assessed the predictive power of the structural model for the hold-out sample. The PLS-predict feature in Smart PLS [71] is run by six-fold cross-validation and 10 algorithm repetitions [78]. All COIN indicators meet the predictive relevance (Q2 0.306–0.436 > 0), and their RMSE values from the PLS model show a less-than-linear regression. It is concluded that the structural model has high predictive power to estimate unpredictable reactions.

## 5. Discussion

Some studies explored the external and internal factors that influence the intention to convert from LPG to electricity. This current study, therefore, examines the quantitative correlation between the predictor variables and the intention of LPG consumers' point of view. It is among only a few pieces of research that have assessed the impact of urban convenience on conversion intention. The results of the proposed study showed that integrating TCV and VAM generated a favorable model with both explanatory and predictive power, providing a suitable foundation for future research.

The conversion intention of urban residents is significantly influenced by a product's time-oriented convenience. In the context of social e-commerce, Mamonov and Benbunan-Fich [50] stated that time-saving convenience influences the intention to use a virtual gift store. Another study conducted in the context of switching to reading e-books in Taiwan by Chiang and Chen [32] explained that intention is influenced by time convenience when

reading e-books anywhere and at any time, which is consistent with the findings of this study. Moreover, Shahijan et al. [34] and Shankar and Rishi [49] stressed that service convenience positively influenced behavior intention. However, on the contrary, in a condition of limited information about product quality [50] or well-known products [52], convenience had an insignificant impact on intention.

Nonetheless, time-oriented convenience strongly affects the perceived consumption value of EKA itself. Therefore, perception plays an important role in escalating the conversion intention from an LPG-fueled stove to electric power as cooking energy. This finding supports García-Fernández's opinion that more time and physical effort would detract from the client's perceived value [33]. In part, Shahijan et al. [34] and Amy Wong [54] discovered the importance of service convenience in influencing perceived value. It implies that marketers should focus on how to build the perception of consumption value based on the value proposition that consumers seek (urban convenience) so that they are more interested in converting their cooking energy.

Induction stoves and air fryers are still unfamiliar to Indonesian citizens, even in urban areas, because of the massive conversion program from kerosene to LPG in 2007 [12,13,79]. Searching for information about a new product and a willingness to seek out the newest technology received the highest score for consumption value perceived by urban residents. Therefore, epistemic value is the strongest value to motivate conversion, followed by conditional value and functional value. Accessibility to an information source about the advantages of EKA in any aspect from the customer's point of view, such as EKA price comparisons, operational cooking cost reduction, and how it would change their daily routine.

Conditional value is reflected by the availability of subsidized electric bills. Interestingly, consumers admittedly do not know exactly how much their electric bill and LPG expense are (Appendix A, Table A1) because of misleading information [12]. In the interviews we conducted, the respondents expected discounts or government subsidies because they perceived electricity as a more expensive commodity [80] as a result of the PLN social marketing advertisement on television and radio prior to the 35 GW project [81,82]. Even still, the limited access to information on the performance and cost efficiency of EKA influences a lower intention to utilize them. This result would be a good prediction that targeting urban residents at 1300 VA or above for converting their cooking energy from LPG to electric power would be more successful than a lower segment of installed capacity. Economic capability affects the purchasing power of consumers to buy more electronic appliances and increase electricity consumption for their life convenience. So, electronic device literacy results in lowering the barriers to switching from LPG-fueled stove to electric kitchen appliances [83].

Functional value is also an extrinsic, basic motivation for using products to support our daily life. The most distinctive urban lifestyle is appreciating time as a scarce resource that is priceless. The results indicate that the time-oriented appeals categorization based on a content analysis of magazine advertisements for urban readers from more than 30 years ago (conducted by Gross and Sheth [36]) is still applicable to the indicators of time-oriented urban convenience. Conversion cost is not a mooring factor in this electricity customer segment that influences the perceived alternative value of electric kitchen appliances to motivate conversion from LPG to electric power for household cooking energy. Future research could find a different result at lower social class and installed capacity segments of customers. The research findings suggest that perceptions about consumption values expected to be fulfilled by using the product are crucial for encouraging electric kitchen appliances as an alternative cooking energy converter. These results emphasize that time-oriented features embedded in urban convenience as an extrinsic benefit that influence consumption values perceived. The time and effort expended as a one-time conversion cost at the beginning of the conversion process do not hinder conversion intentions.

## 6. Conclusions

This study investigates the role of the effects of urban convenience on conversion intention and the perceived alternative value in mitigating urban convenience effects on conversion intention regarding a change from LPG-fueled stoves to electric kitchen appliances. It also seeks to find out the effect of the mooring effect of the conversion cost barriers on conversion intention.

The findings reveal that PAV, as formed by the consumption values of the alternative product, plays a major role in determining the conversion intention of cooking energy. Moreover, the study finds that PAV partially mediates the relationship between urban convenience and COIN. However, the conversion cost is found to have no significant moderating effect on PAV and COIN.

The study has some implications for the management of PLN as a state-owned electricity producer and the Ministry of Energy and Minerals Resources. As confirmed, conversion intention is determined by perceived alternative value and urban convenience. For this reason, they should focus on introducing electricity as part of the urban lifestyle. Synergistic co-operation in shaping a new urban lifestyle by including the "electrifying your lifestyle" message through the use of electrical appliances for the convenience of daily life. For example, (a) electric cookware manufacturers collaborate with artists in dramas and reality shows; (b) a school education curriculum that introduces the advantages of electricity for life and project-based assignments for students in designing public service advertisements; (c) the Ministry of Energy and Mineral Resources needs to increase the involvement of influencers and vloggers to raise the theme of an electrifying lifestyle. Eliminating the perception that electricity cost is higher than LPG [80] would minimize the subsidized price as an obstacle to conversion intention. Therefore, the program of reducing LPG consumption and increasing electricity for cooking energy as part of a new urban lifestyle targeting the segment of 1300 VA and above would be more successful than segment 450–900 VA.

From this study, the urban convenience and perception of EKA consumption value (PAV) indicate their impact on conversion intention by integrating the theory of consumption value to a value-based adoption model. Additionally, time-oriented advertisement [36] applied as urban convenience indicators positively influences perceived alternative value and conversion intention. Thus, EKA manufacturers should consider time-oriented convenience impact on perceived EKA consumption value by transforming it into product features, advertisement messages, and other communication materials.

Consumption value is uniquely perceived depending on time and person [26]. The result of this study might have been different if it had been applied to the lower segment. Future research should be undertaken in the lower segment and consider a wider range of areas, including remote areas (nongrid system) and rural areas, in which the reliability of electricity supply is an issue. It should also consider before and after experiencing electric kitchen appliances.

Finally, it would be fruitful to study other aspects affecting conversion intention. When compared to installing gas tube regulators, EKA is much easier. The perceived ease of use might have a significant impact on conversion intention as rural Indonesians had experienced switching from kerosene to LPG-fueled stoves [79]. The value of money invested in new kitchen utensils affects the usefulness of electric kitchen appliances. Even though there was news of an explosion of gas cylinders, there was also news of a house fire due to an electric short circuit that affected the perceived safety of electrical installations in the house. On the other hand, external factors might prohibit them from switching. How often and how long the average customer experiences an interruption (available 24/7) and reliability of electricity is one of the conditional factors that is impactful too. In addition, easy access to subsidized LPG supplies due to lower operational costs might also hinder conversion intentions. Therefore, comparing those customers inside of the Java Bali grid vs. those outside the Java Bali grid within isolated islands and shortages of LPG and/or electricity would come out with a different response.

**Author Contributions:** Conceptualization, H.H.-M., A.S., F.R. and A.Y.; methodology, H.H.-M., A.S., F.R. and A.Y.; software, H.H.-M.; validation, A.S., F.R. and A.Y.; formal analysis, H.H.-M.; investigation, A.S., F.R. and A.Y.; resources, H.H.-M.; data curation, H.H.-M.; writing—original draft preparation, H.H.-M.; writing—review and editing, H.H.-M., D.F.H. and H.N.; visualization, H.H.-M.; supervision, H.H.-M., D.F.H. and H.N.; project administration, H.H.-M.; funding acquisition, H.N. All authors have read and agreed to the published version of the manuscript.

**Funding:** This research was partly funded by PT Perusahaan Listrik Negara (Persero) or PT PLN (Persero), a state owned electricity company in Indonesia.

**Institutional Review Board Statement:** Not applicable.

**Informed Consent Statement:** Not applicable.

**Data Availability Statement:** Data available on request.

**Conflicts of Interest:** The authors declare no conflict of interest.

## Appendix A

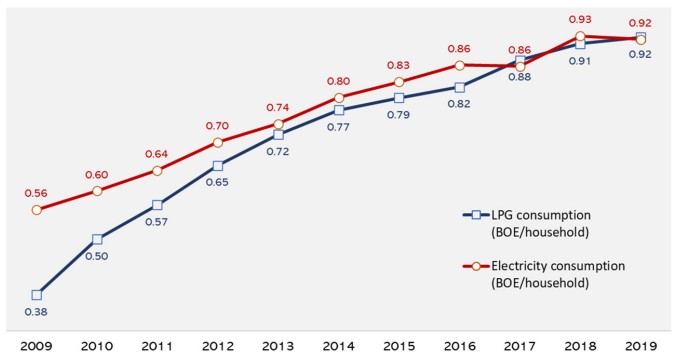

**Figure A1.** Household consumption: LPG vs. electricity. Source: [84].

**Table A1.** Energy consumption pattern.

| Questions | Mean | Standard Deviation | Excess Kurtosis | Skewness | Frequency |
|---|---|---|---|---|---|
| In the last 3 months, my electricity consumption tends to be ... | 1.753 | 0.443 | −0.488 | −1 | decrease 1; steady 144; increase 49 |
| In the last 3 months, my LPG consumption for cooking tends to be ... | 1.954 | 0.384 | 3.704 | −0.46 | decrease 10; steady 165; increase 19 |
| In the last 3 months, I consumed LPG more than electric power for my cooking energy | 3.119 | 1.24 | −1.014 | 0.018 | Strongly agree 34; agree 41; neutral 52; disagree 48; Strongly dissagre 19 |

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
