# Peer review of "Assessing the Impact of Urban Lifestyle and Consumption Values on Conversion Intention: A Study towards Energy Sustainability"

_sustainability, doi:10.3390/su15086549_

Round 1

Reviewer 1 Report

REFEREE'S REPORT ON

"Assessing the Impact of Urban Lifestyle and Consumption Values on Electricity Stove Conversion Intention in Jakarta: A Study towards Energy Sustainability"

Comments: The authors studied “the impact of urban lifestyle and consumption values on electricity stove conversion intention” in Jakarta. This paper needs improvement as has been listed below:

1.      Abstract

Ø  The policy proposal of the study's originality and importance should be written in this section.

Ø  A brief description of the model of the study should be written.

2.      Introduction

Ø  Theoretical explanations regarding the impact of urban lifestyle and consumption values on electricity stove conversion intention are insufficient.

Ø  The study's importance, purpose and theoretical framework should be discussed in detail.

Ø  Reference should be made below Figure 1. Use the "source" header. Also, figure 1 has not been sufficiently discussed.

Ø  You don’t need to write the results of the study in the introduction section.

3.      Literature

Ø  It is very important to approach literature studies critically. Also more recent and current studies should be added to these citations. Literature research on the relevant subject is not sufficient. I suggest creating a literature table.

Ø  The difference of the study from the literature and its contribution to the literature should be explained under this title.

4.      Data and Methodology

Ø  Analyzes and the findings have not been adequately discussed.

Ø  You should calculate statistical values such as kurtosis, skewness and JB in descriptive of statistics at the Table 3.

5.      Conclusion

Ø  The conclusion and recommendations section is very successful. But the results obtained should be discussed. The study should be compared with the literature. The contribution of the study to the literature and which studies it contradicts and supports should be given critically, along with its justifications. Also plenty of policy proposals should be presented.

Author Response

Dear Reviewer, 

Thank you for your feedback on our manuscript. Please find attached the revised version, which includes the addressed comments from your review. We sincerely appreciate your valuable comments and believe that they have greatly improved the quality of our work.

If you have any further comments or suggestions, please do not hesitate to let us know.

Best regards, 

Reviewer 2 Report

1.      The structure of the paper is not reasonable. The section 2.2 should be placed in the Method section.

2.      The originality of the paper needs to be further clarified. The present form does not have sufficient results to justify the novelty of a high quality journal paper.

3.      -The results should be further elaborated to show how they could be used for the real applications.

4.      -The paper should be written from the international perspective rather than focusing on the issues of one country.

5.      The investigation is complicated and tedious, how do you guarantee the quality of the investigation? Maybe you should talk about more details in your Method section.

6.      More recent literature is needed to broaden the view of readers.

Author Response

Dear Reviewer. 

Thank you for your feedback on our manuscript. Please find attached the revised version, which includes the addressed comments from your review. We sincerely appreciate your valuable comments and believe that they have greatly improved the quality of our work.

If you have any further comments or suggestions, please do not hesitate to let us know.

Best regards,

Round 2

Reviewer 2 Report

This paper has been greatly improved and I have no further questions